# Influence of the Dielectric Coating of the Outer Side of the Cathode in the Anode–Cathode Pairs of a Molecular Electronic Sensitive Element on the Conversion Coefficient

**DOI:** 10.3390/mi13030360

**Published:** 2022-02-24

**Authors:** Alexander Bugaev, Victoria Agafonova, Ivan Egorov, Ekaterina Agafonova, Svetlana Avdyukhina

**Affiliations:** 1Moscow Institute of Physics and Technology, Dolgoprudny 141701, Moscow Region, Russia; bugaev@cplire.ru (A.B.); agafonova.va@mipt.ru (V.A.); egorov.iv@mipt.ru (I.E.); avdysvetlana@mail.ru (S.A.); 2Special Machinery Department, Bauman Moscow State Technical University, Moscow 105005, Russia; 3R-Sensors, LLC, Dolgoprudny 141700, Moscow Region, Russia

**Keywords:** electrochemical sensor, molecular-electronic transfer, microelectrodes, diffusion, electrolyte, sensitivity, laser micromachining

## Abstract

Molecular electronic sensors of motion parameters use miniature electrochemical cells as a sensitive element, in which the interelectrode current is sensitive to external mechanical influences. New approaches for creating conversion elements are based on precision methods of micromachining materials. The use of new technologies has opened up the possibility of creating sensitive elements with configurations that have not been previously studied, and for which there is no clear understanding of the regularities that determine the output parameters depending on the geometry of the conversion elements. This work studies the influence of the dielectric coating on the surface of the cathodes on the conversion coefficient. The transforming structure has been made from three plates. The outer plates were an anode–cathode electrode pair. The middle plate served as a separator between the pairs of electrodes. It was found that an insulating layer on the side of the cathode facing away from the adjacent anode allows the conversion factor to be doubled. This result is applicable for a wide class of conversion elements made with microelectronic technologies, as well as structures made of mesh electrodes.

## 1. Introduction

Molecular electronic sensors of motion parameters and wave fields use miniature electrochemical cells as a sensitive element, in which the interelectrode current is very sensitive to external mechanical impact [1,2,3,4]. Highly sensitive motion sensors, such as seismometers, accelerometers and angular velocity sensors, have been created on the basis of such cells [5,6,7,8,9,10,11,12,13]. For some applications, such as seismic exploration, it is about creating hundreds of thousands or even millions of such devices per year [14,15,16].

The success achieved in the technical characteristics and understanding of the broad commercial prospects of this direction, prompted researchers from around the world to develop technological solutions based on using precision methods of microfabrication, such as microelectronic fabrication and/or laser drilling [4,17,18,19,20], which reduce the scatter of parameters of sensitive elements, reduce their cost and prepare them for mass production. The first results related to the use of microelectronic technologies were published in 2012–2013 by research groups from Moscow Institute of Physics and Technology [21], Arizona State University [22] and the Institute of Electronics CAS [3,23,24].

In this case, the technological processes of oxidation in a diffusion furnace [21], explosive photolithography and electron–beam deposition [21,22], laser cutting [3,21,24], low-pressure chemical vapor deposited (LPCVD) [22], plasma-enhanced chemical vapor deposited (PECVD) [22], focused ion beam (FIB) [22], deep reactive ion etching (DRIE) [3], [22] were used. Over the next 10 years, these technologies were significantly developed and, nowadays, we can discuss the presence of several tested different solutions, which make it possible to successfully manufacture conversion elements with high sensitivity and good repeatability [4,17,18,19,20].

At the same time, sensing elements configurations produced using precision microprocessing differ from previously used grid electrode designs, and, until recently, they were not studied due to the lack of methods for their creation.

As a result, advances in the field of technology have not yet been adequately supported by progress in the study of physical processes that determine the output parameters of the final devices. Consequently, technologists do not have an unambiguous understanding of what characteristics are required, and, primarily, what geometry of the sensitive element and individual parts of its structure allow us to obtain its lowest self-noise and highest sensitivity.

Regardless of the specific implementation, the design of the electrochemical sensing element contains a system of electrodes placed in miniature channels filled with a highly concentrated electrolyte solution, between which a potential difference is applied. The electrolyte solution for this type of sensors is both the working medium of the sensing element and the inertial mass. External mechanical signals create inertia forces acting on the liquid and ensure its flow through these channels. All practical cells comprise two pairs of electrodes placed in the channels in the order AC–CA (anode–cathode–cathode–anode), so that for any direction of the inertial force, the liquid moves from the anode to the cathode in one pair of the electrodes and from the cathode to the anode in the other pair. At the same time, for one pair of electrodes, the liquid flow accelerates the delivery of ions formed at the anode to the cathode and, in the other pair, it slows it down. Accordingly, the cathode electric current changes in each pair. The difference in the cathode currents is the output signal of the sensing element. The advantage of the described principle of measuring the mechanical movements is a high conversion coefficient, i.e., the ability to obtain high-level primary electrical signals, significantly higher than the level of self-noise of subsequent stages without pre-amplification, even for weak mechanical influences. Accordingly, a high conversion factor usually allows a higher signal-to-noise ratio to be obtained even when registering weak signals.

In this paper, a special case of the conversion element geometry was studied, which arouses interest because it allows us to ensure good repeatability of the output parameters. Each pair of electrodes is formed on two sides of one non-conductive plate. The specified plate has through holes to ensure the flow of the working fluid. The final four-electrode package is obtained by connecting the two indicated plates to each other through a special dielectric separator. The advantage of this design is that the most significant effect on the characteristics of the conversion element is exerted precisely by the mutual arrangement of the anode and cathode, which, with this approach, can be ensured with a sufficiently high accuracy.

The objective of the study performed in this work was to study the effect of a dielectric coating deposited on the surface of the plates on which anode–cathode pairs are formed from the cathode sensor conversion factor. This work studied the specific question that had not been studied yet, which is to investigate the effect of a dielectric coating applied on the surface of the cathodes on the sensor conversion factor. The study was carried out by experimental methods; therefore, different samples were made, with and without such a coating. The measurements carried out show that the specified dielectric coating provides a higher conversion coefficient at a close value of the background current, i.e., the current flowing between the electrodes in case of stationary liquid. In addition, the manufactured samples differed in the distance between the holes for the flow of the working fluid and the number of such holes. It was found that an increase in the number of holes for the flow of liquid through the sensing element leads to an increase in the conversion factor.

## 2. Materials and Methods

The design of the conversion element of the studied type is shown in Figure 1. The conversion element can be represented as a structure formed by three plates. The central plate has 1 relatively large through hole with a diameter of 7 mm, the outer plates have many small holes with a diameter of 20 μm and the electrodes produced on the opposite sides of these plates, with the anodes located outside the package and the cathodes located on the inner side of the plates.

The difference between the two studied configurations is that in Option 1 (Figure 1, left), the side of the cathodes parallel to the surface of the plates is covered with a dielectric layer, and, in Option 2 (Figure 1, right), it is left open.

The experimental samples were made using thick film technology [25,26,27] and laser precision microfabrication [28,29]. The diagram of the upper and lower plates manufacturing process is presented in Figure 2.

The original material for the manufacture of the plates containing electrodes was a mount from corundum BK-100-type ceramics of 150 µm thickness produced by JCS Polycor (Kineshma, Russia) [30]. Initially, a notch of ~30 µm depth was engraved on the substrate surface. Then, the indicated grooves were filled with conductive polymer paste PPl (paste platinum), the conductive component of which was 100% platinum, produced by DEPA (Zelenograd, Moscow Region, Russia) [31]. Annealing of the paste was carried out at a temperature of 850 °C for 2 h 30 min. 

The next step was to form a dielectric coating on the inside of the plate. For this, a dielectric PD paste also produced by DEPA (Zelenograd, Moscow Region, Russia) [31], which was applied over the conductive layer formed in the previous stage and then annealed. Such a dielectric coating was created only for samples whose design was consistent with Option 1. Next, a conductive layer was created using conductive polymer paste, PP, on the back of the workpiece, and through holes were made using laser irradiation from the anode side. The diameter of the through holes was measured using a measuring microscope and was 25 ± 2 µm on the anode side and 21 ± 2 µm on the cathode side.

The number of holes was chosen to be the maximum quantity, provided that the integrity of the dielectric coating was maintained. Initially, on test samples, it was found that when the distance between the centers of the holes decreases to less than 100 microns, areas of delamination from the substrate appear in the coating. Therefore, samples with a smaller distance between the holes were not made and their characteristics were not studied. The size of the working area of the transforming element was chosen as 5 × 5 mm. A total of 3 sets of the samples were produced with a different number of holes, namely 2500, 1600 and 1024. Accordingly, the distances between the centers of the holes for the samples were 100, 120 and 150 microns. Contacts for all the layers were made by gluing a platinum wire with a diameter of 100 µm using a specified conductive paste to the conductive layers, followed by annealing. Since the manufacturing process of the conductive contacts is obvious, it is not shown in Figure 2.

The central plate of the conversion element was cut from corundum BK-100-type ceramic mounts with a thickness of 150 µm. The plates were glued together with a dielectric paste (PD) (Zelenograd, Moscow Region, Russia) [31], after which the prepared package was baked again. The samples can be seen in Figure 3.

Thus, the difference between the two types of conversion elements is that for the ‘Option 1’ structure, in which the sides of the cathodes opposite to the adjacent anode remained open, while, for the ‘Option 2’ structure, they were covered with a dielectric layer.

The conversion coefficient of the conversion elements was studied in the working band from 0.32 to 500 Hz using a special testing device containing an electrodynamic actuator. For the installation in this device, the conversion element made, as described above, was enclosed in a special holder. The holder was a plastic tablet, in the central part of which there was a conversion element made using a 3D printer, which significantly reduced production costs and time in a laboratory, since working with such a device does not require special conditions for the equipment operation and complex skills.

The process of making the holder is shown in Figure 4a, a 3D model of the transforming element placed in the holder is shown in Figure 4b, and its actual view can be seen in Figure 4c.

Next, the conversion element in the holder was installed in the body of the device for testing. A diagram of the test device and its view are shown in Figure 5a,b, respectively. To test the conversion element, a coil (2) of an electromagnetic actuator was placed on the housing (1), interacting with a magnet (3) fixed on one of the membranes (4). Two diaphragms (4) limit a volume containing a plastic holder (5) with a conversion element (6) and filled with working fluid (7). The working fluid was an aqueous solution of lithium iodide of a concentration of 4 M, with the addition of the active component of 0.03 M of iodine.

To check and compare the cells of different configurations, the amplitude-frequency response of the sensing elements was obtained by applying an electrical current into the excitation coil, which impacted a magnet fixed to the flexible membrane as described above. The impact was performed by harmonic current variations in the coil in the range of 0.32–500 Hz. The ratio of the output current to the input current in the coil, $W_{I}$, is measured for a given frequency range. According to [32], the amplitude characteristic measured this way coincides with the accuracy of the coefficient calibration coil motor constant with the sensor acceleration amplitude characteristic $W_{A}$.
(1)$$W_{A}=K_{A}W_{I}$$

Since practically used sensors of this type most often have an output that is proportional to the velocity and not to the acceleration, additional results were reduced to an output signal proportional to the speed by multiplying by a factor of 2πf:(2)$$W=2\pi fW_{I}$$
where f is the exposure frequency.

## 3. Results

Six samples of ‘Option 1’ and six samples of ‘Option 2’ were made and tested, differing in the number of holes and the pitch between the holes. The results of measuring the frequency dependence, *W*(*f*), are presented below in Figure 6. Sample characteristics, such as the sample ID, configuration type (as described above), number of the holes, and thickness of the central plate, are presented below in columns 1 to 4 in Table 1. Additionally, the measured parameters, such as the background current, conversion factor $W\left(8\right)$ at 8 Hz (as Figure 6 shows, it usually corresponds to the maximum value for the amplitude vs. frequency responses), and conversion factor $W\left(0.32\right)$ at 0.32 Hz (lowest measured frequencies), are summarized in Table 1, in columns 5 to 7.

The purpose of the subsequent analysis of the results obtained is to isolate the effect of the dielectric coating on the reverse side of the cathode on the conversion factor. 

First, note that the influence of the geometry of individual electrodes, including the presence or absence of a dielectric coating on the cathodes, is significant only at sufficiently high frequencies, namely, under the condition of $f>f_{D},$ where f is the input signal frequency and fD~D2πR2. R is the radius of the channel in which the working liquid flows. At lower frequencies, the dominant signal conversion mechanism is diffusion to the electrodes from the volume of the working channel of the active ions transported there by the liquid flow. For the considered geometries, as described above, the radius of the channel is about 10 µm, and at $D=2\cdot 10^{-9}\;m^{2}/s$, the frequency is $f_{D}\sim \;3\;\mathrm{Hz}$. More specifically, consider the frequency f=8 Hz, corresponding, moreover, to the maximum conversion coefficient of the experimental samples under study. 

Additionally, in order to exclude the influence on the analysis results of the non-identity of such parameters as the constant of interaction between the magnet and the coil in the experimental calibration device shown in Figure 6, the spread in the concentration of the active ions of working solutions, errors in creating the interelectrode distance, use the value of the conversion coefficient at the selected frequency, 8 Hz, normalized to the conversion factor measured at the lowest frequency used in calibration 0.32 Hz: Wn=W(8|)W0.32.  The calculation results for the normalized conversion factor are presented in the last column in Table 1.

A comparison of the conversion factors for Option 1 and Option 2 shows that the conversion factor of the conversion elements for Option 1 is always higher than for Option 2, regardless of the total number of holes made in the conversion element. Thus, the measurements performed show that the background currents for the studied samples differ by no more than 20%, while the conversion coefficient for the ‘Option 1’ sample is approximately twice as high.

Figure 7 illustrates the physical explanation of the observed effect of the dielectric coating. The left side of Figure 7 corresponds to the case of the absence of a dielectric coating; the right side corresponds to the sensing element with a dielectric coating. The arrows in the figure schematically show the trajectories along which the movement of ions occurs, created as a result of electrochemical reactions at the anode to the cathode. It can be seen from the presented diagram that, at the moment when the liquid flow is directed from left to right, as shown in Figure, in most of the space, the hydrodynamic flow carries ions towards the cathode contributes, for example, to an increase in the interelectrode current. At the same time, for a sensitive element without a dielectric coating, there is an area of space outlined in red, in which the ion flow and the hydrodynamic flow of the liquid are directed mainly in opposite directions. Accordingly, the liquid flow carries the ions away from the cathode, which reduces the interelectrode current. The overall manifestation of the effect is a drop in the conversion factor compared to the case when the reverse side of the cathode is covered with a dielectric (right side of Figure 7).

Another important observation is that the maximum conversion factor for samples containing a dielectric layer is always shifted towards higher frequencies. It is close to 8 Hz, comparing to 3 Hz for samples without a dielectric coating.

A comparison of the results obtained for configurations with different numbers of holes in the channels shows that an increase in the number of holes leads to an increase in the normalized conversion factor and background current for both Option 1 and Option 2, while the location of the maximum at frequency dependence does not depend on the number of holes. The conclusion can be drawn that the dielectric coating does not collapse with a decrease in the distance between the holes, and the increase in the conversion coefficient is explained by the increase in the number of channels for the flow of the working fluid.

## 4. Discussion

Applying an insulating layer to the side of the cathode facing away from the adjacent anode allows for the conversion factor to be approximately doubled without a noticeable increase in the current consumption. Quantitatively, the normalized conversion factor increases 2.81 ± 0.13, 2.0 ± 0.3 and 2.0 ± 0.6 times, respectively, for the configurations containing 2500, 1600 and 1024 holes.

In particular, this result shows that the deposition of a dielectric on one of the sides of the cathode can increase the sensitivity of classical transforming structures made of mesh electrodes, as well as modern conversion elements, such as those presented in [19,33,34]. In the design proposed in [19], which was further improved in [33,34], the anode is located on one side of the silicon wafer, and the cathode is partially located on the same side of the wafer, partially on the sides of the holes connecting the sides of the wafer, and most of it is on the opposite side of the plate. The result obtained in this work shows that the conversion coefficient for such a design can be significantly increased if the reverse side of the plates is isolated. The verification of this assumption is essential as it may provide a very promising method for improving the efficiency of signal conversion in electrochemical sensors using this design modification.

In addition, taking into account the results of this work, it would be very interesting to study, in the future, the effect of the partial coverage of the cathode of the conversion element on the self-noise of the converter. The point is that, as shown, for example, in [35], at high frequencies, the self-noise of the electrochemical converter decreases with increasing cathode impedance. Accordingly, it can be expected that the application of a dielectric coating simultaneously increases the impedance and the conversion factor, which can qualitatively improve the signal-to-noise ratio of sensors at high frequencies, which is critically important [36] for a number of applications, such as seismic exploration and hydroacoustics.

For a specific method of manufacturing a sensitive element based on the thick film technology and laser processing used in this work, it can be stated that the technology allows for a decrease in the distance between the holes in the conversion element to at least 100 μm, and at the same time a positive effect of the conversion coefficient is completely preserved, which indicates that the integrity of the coating is preserved after the holes are made by the laser drilling method. The best results were obtained with a sensing element containing 2500 holes with a hole center distance of 100 µm.

The obtained result can be expected to prove to be useful for the creation of highly sensitive electrochemical conversion elements, regardless of the technological process used to create them.

## Figures and Tables

**Figure 1 micromachines-13-00360-f001:**
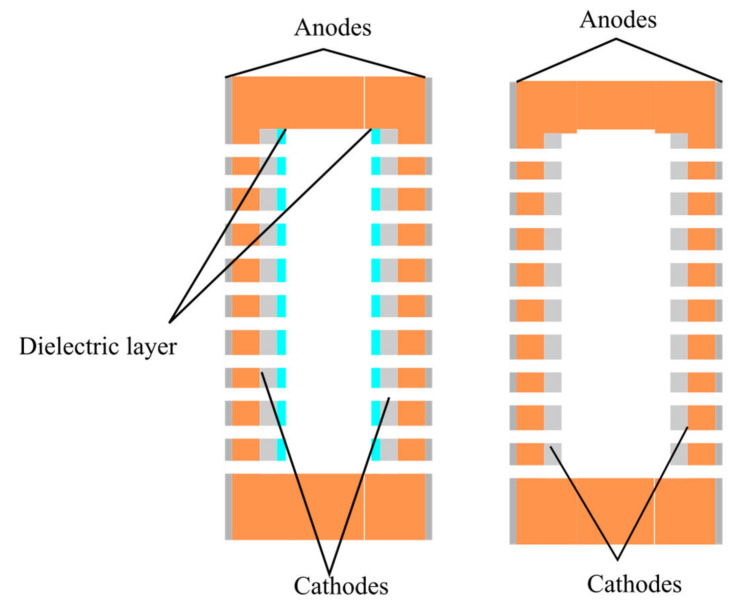
Two configurations of the signal converting cell. The left one shows Option 1, the configuration with a dielectric layer; the right one shows Option 2, the configuration without a dielectric layer.

**Figure 2 micromachines-13-00360-f002:**
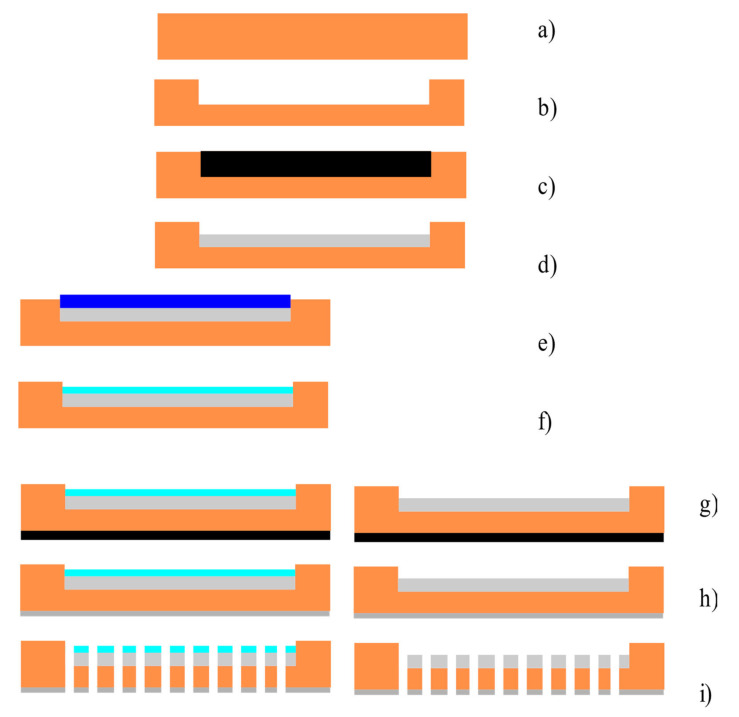
The diagram of the manufacturing process for the plates of the electrochemical cell containing the electrodes. (**a**) Corundum substrate; (**b**) laser micro-milling bath for the platinum containing polymer pastes; (**c**) application of the platinum containing polymer paste; (**d**) annealing; (**e**) application of the non-conductive polymer paste; (**f**) annealing of the non-conductive polymer paste; (**g**) application of the platinum containing polymer paste on the opposite side of the corundum substrate; (**h**) annealing; (**i**) laser drilling.

**Figure 3 micromachines-13-00360-f003:**
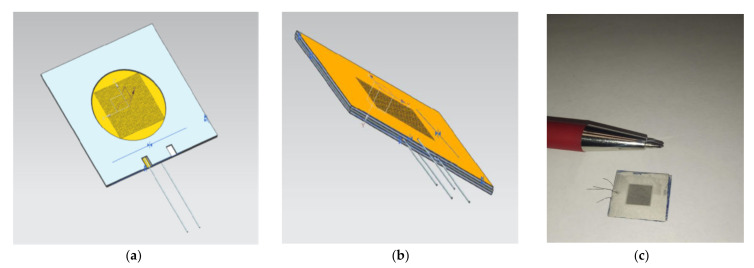
The conversion element design (**a**) partially assembled element—outer plate with a dividing plate; (**b**) a schematic view of the package assembly; (**c**) the conversion element.

**Figure 4 micromachines-13-00360-f004:**
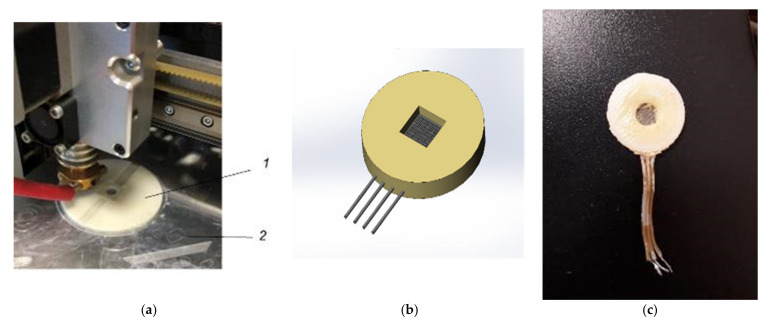
Manufacturing of the holder for the conversion element (**a**) of the 3D body; (**b**) a model of the electrochemical conversion element; (**c**) the finished conversion element in the holder.

**Figure 5 micromachines-13-00360-f005:**
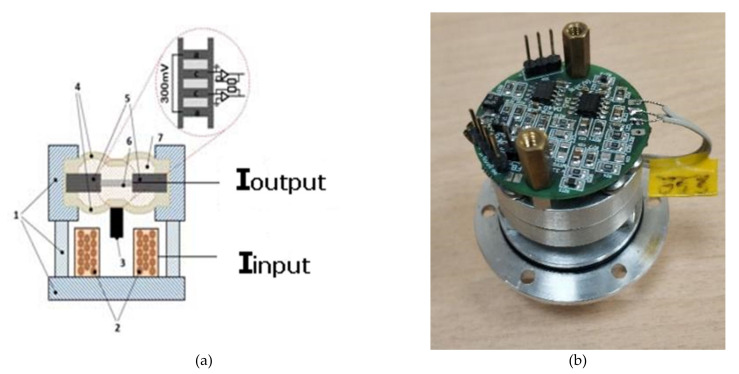
Sample assembled with the electrodynamic calibrator. (**a**) diagram of the device; (**b**) actual view. 1—sensor body with the coil holder; 2—coil; 3—magnet; 4—membranes; 5—holder of the conversion element; 6—conversion element; 7—electrolyte solution.

**Figure 6 micromachines-13-00360-f006:**
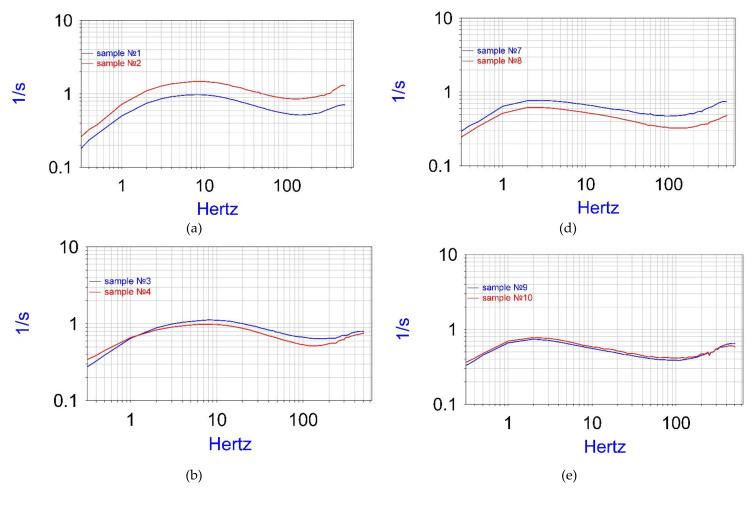
Amplitude vs. frequency curves for the experimental samples. **Left** side—configuration 1. **Right** side—configuration 2. From **top** to **bottom** (**top**): 2500 holes, (**middle**) 1600 holes, (**bottom**) 1024 holes. (**a**) samples 1 and 2; (**b**) samples 3 and 4; (**c**) samples 5 and 6; (**d**) samples 7 and 8; (**e**) samples 9 and 10; (**f**) samples 11 and 12.

**Figure 7 micromachines-13-00360-f007:**
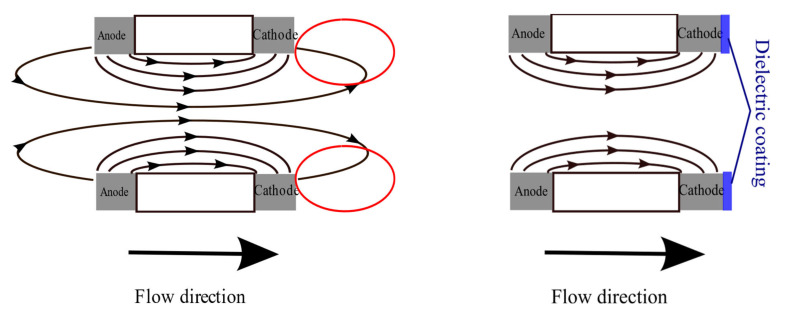
Ion transport in the absence (**left** side) and presence of a dielectric coating (**right** side). The lines show the movement of ions from the anode to the cathode due to diffusion. Circled in red are the areas of space in which the hydrodynamic transfer of ions leads to a decrease in the output signal.

**Table 1 micromachines-13-00360-t001:** Summarized parameters and of the experimental samples.

Sample ID	Option	Number of Holes	Central Plate Thickness, µm	Average Background Current, µA	Conversion Coefficient Measured at 8 Hz *W*(8)	Conversion Coefficient Measured at 0.32 Hz *W*(0.32)	Normalized Conversion Coefficient *W*(8)/*W*(0.32)
1	1	2500	150	84	0.98	0.18	5.44
2	1	2500	150	89	1.42	0.26	5.46
3	1	1600	150	59	1.13	0.29	3.88
4	1	1600	150	60	0.99	0.32	3.09
5	1	1024	150	25	0.36	0.15	2.40
6	1	1024	150	26	0.38	0.14	2.71
7	2	2500	150	59	0.69	0.37	1.86
8	2	2500	150	61	0.55	0.28	1.96
9	2	1600	150	62	0.59	0.33	1.79
10	2	1600	150	59	0.60	0.36	1.69
11	2	1024	150	29	0.30	0.25	1.20
12	2	1024	150	31	0.18	0.24	0.75

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
