# Peer review of "Influence of the Dielectric Coating of the Outer Side of the Cathode in the Anode–Cathode Pairs of a Molecular Electronic Sensitive Element on the Conversion Coefficient"

_micromachines, 2022, doi:10.3390/mi13030360_

Round 1

Reviewer 1 Report

Introduction include vague terms as “new technologies” and the objective is not clear.

Main idea as conversion coefficient is not defined.

The sensing process is not well defined.

Figures are not well labeled, legend have different formats, axis different scales…

Diffusion units are strange.

Table 1 lacks information “option 2”, ID sample is not relevant, …

Equation 1 and 2 are not well introduced.

There is no blank experiment or statistics.

Author Response

Authors greatfully thank  you for the analysis and very valueable  suggestions and comments.

We significant revised the manuscript. You can find the detailes in the attached document.

Besides we asked a professional translator to check and improve the English.

Sincerely Yours, Authors

Reviewer 2 Report

This manuscript studies the effect of the dielectric coating of the outer side of the cathode in the anode-cathode pairs of a molecular-electronic sensitive element on the conversion coefficient. The manuscript has clear research purpose and complete structure. However, it is not so innovative, and the results and discussions are too simple to be practical. My comments and suggestions are as follows. 1) Please give more details of the technical principles, including the basic idea of design differs from existing technology. 2) The introduction also needs to analyze the existing technology level. 3) The laws and principles reflected by the experimental results. How to apply? 4) Please give the reasons for selecting the number of holes in the experiment. 5) Could the quantitative index be given in the conclusion?

Author Response

Dear Reviewer,

we thank you for accurate reading and vey useful and professional comments and suggestions.

We used ones for the revision. You can find a point by point answers in the attached document.

Besides we asked a translator to check and correct English over the manuscript.

Best regards,

Kate Agafonova. Corresponding aurthor.

Round 2

Reviewer 1 Report

In this version there are substantial improvements in the introduction and explanation of the physical principle of operation. However, more effort could be put into selecting figures that do not duplicate information, explain the impact of the results for other groups or applications, and envision future lines of research.

Author Response

Dear Reviewer,

we thank you for your efforts and very valuable comments and suggestions.

The following are our response to your comments:

 However, more effort could be put into selecting figures that do not duplicate information...

We modified the figures 7 to avoid duplicating information presented in Table 1.

Specifically, we deleted information about back ground current since it is already presented in Table 1.

explain the impact of the results for other groups or applications, and envision future lines of research.

 Specifically, in the design proposed in [33], which was further improved in [34, 35], the anode is located on one side of the silicon wafer, and the cathode is partially located on the same side of the wafer, partially on the sides of the holes connecting the sides of the wafer, and most of it is on the opposite side of the plate. The result obtained in this work shows that the conversion coefficient for such a design can be significantly increased if the reverse side of the plates is isolated. Verification of this assumption is essential as it may provide a very promising method for improving the efficiency of signal conversion in electrochemical sensors using this design modification.

In addition, taking into account the results of this work, it would be very interesting to study in future the effect of partial coverage of the cathode of the conversion element on the self noise of the converter. The point is that, as shown, for example, in [36], at high frequencies, the self noise of the electrochemical converter decreases with increasing cathode impedance. Accordingly, it can be expected that the application of a dielectric coating will simultaneously increase the impedance and increase the conversion factor, which can qualitatively improve the signal-to-noise ratio of sensors at high frequencies, which is critically important [37] for a number of applications, like seismic exploration, hydroacoustic, etc.

The corresponding information is added  into the manuscript (lines 268-284).

Sincerely Yours, Authors.

Reviewer 2 Report

If you can give a point-to-point modification instruction file, it will help me understand.

Author Response

Dear Reviewer,

thanks for your comments and suggestions.

While answering your previous version of the comments and suggestions we have prepared a point by point file with our answers. For some technical problems, it is quite possible that you did nor receive this file. 

Sorry for this. Here we attach this document for your consideration. 

Hopefully, it helps you to understand the manuscript modifications we made based on your comments and suggestions.

Sincerely Yours, Authors.

Round 3
